# Lifestyle Variables Such as Daily Internet Use, as Promising Protective Factors against Cognitive Impairment in Patients with Subjective Memory Complaints. Preliminary Results

**DOI:** 10.3390/jpm11121366

**Published:** 2021-12-14

**Authors:** Hernán Ramos, Mónica Alacreu, María Dolores Guerrero, Rafael Sánchez, Lucrecia Moreno

**Affiliations:** 1Cátedra DeCo MICOF-CEU UCH, Universidad Cardenal Herrera-CEU, 46115 Valencia, Spain; ramgarher@alumnos.uchceu.es (H.R.); monica.alacreu@uchceu.es (M.A.); ma_dolores.guerrero@uchceu.es (M.D.G.); rsanchezroy@yahoo.es (R.S.); 2Department of Pharmacy, Universidad Cardenal Herrera-CEU, CEU Universities, 46115 Valencia, Spain; 3Embedded Systems and Artificial Intelligence Group, Universidad Cardenal Herrera-CEU, CEU Universities, 46115 Valencia, Spain; 4Neurology Service, Arnau de Vilanova Hospital, 46015 Valencia, Spain

**Keywords:** cognitive impairment screening, cognitive reserve, subjective memory complaints, internet, television, reading, marital status, sleep

## Abstract

Subjective memory complaints (SMCs) may be important markers in the prediction of cognitive deterioration. The aim of this study was to find associations between individual lifestyle factors, which may contribute to cognitive impairment (CI) in people with SMCs and to conduct a literature review on the relationship between internet use and CI in subjects over 50 years old, as a related factor. This was a case-controlled study that included 497 subjects aged over 50 years with SMCs who were recruited from 19 community pharmacies. Three screening tests were used to detect possible CIs, and individuals with at least one test result compatible with a CI were referred to primary care for evaluation. Having self-referred SMC increased the odds of obtaining scores compatible with CI and this factor was significantly related to having feelings of depression (OR = 2.24, 95% CI [1.34, 3.90]), taking anxiolytics or antidepressants (OR = 1.93, 95% CI [1.23, 3.05]), and being female (OR = 1.83, 95% CI [1.15, 2.88]). Thirty percent of our sample obtained scores compatible with CI. Age over 70 years increased the odds of obtaining scores compatible with CI. A high-level education, reading, and daily internet use were factors associated with a reduced risk of positive scores compatible with CI (37–91%, 7–18%, and 67–86%, respectively), while one extra hour television per day increased the risk by 8–30%. Among others, modifiable lifestyle factors such as reading, and daily internet usage may slow down cognitive decline in patients over 50 with SMCs. Four longitudinal studies and one quasi-experimental study found internet use to be beneficial against CI in patients over 50 years of age.

## 1. Introduction

Alzheimer’s disease is the most common cause of dementia among older adults. It is an irreversible, progressive brain disorder, which slowly destroys memory, thinking skills, and eventually, the ability to carry out simple tasks. Our understanding of the etiology of this disease is poor, although it is known that modifiable lifestyle factors may affect the risk of developing this condition [1]. Therefore, it is important to increase our knowledge of the potential risk factors for Alzheimer’s disease (AD) and to learn to recognize its early symptoms so that it can be detected at earlier stages [2].

Currently, it has been shown that AD is a continuum between the first stages with very few or no symptoms (preclinical AD, MCI) and the most sever phase (dementia) [3]. It is a process in which pathophysiological changes accumulate over several years and culminate in a clinically apparent disease, which then progresses with a gradual decline in cognitive and functional abilities, with no defined boundaries between the different clinical stages [4]. Recently, according to the NIA-AA classification system, AD has been classified into 6 differential stages [3]. Stages 1 and 2 represent those patients who are cognitively unimpaired. Stages 3, 4, 5 and 6, represent individuals with performance in the impaired/abnormal range on the objective cognitive test, and correspond to the different stages of cognitive impairment (CI). Specifically, stage 3 reflects mild cognitive impairment (MCI), while stage 4 reflects mild dementia, stage 5 moderate dementia and stage 6 severe dementia [5].

Stage 2 is related to the so-called subjective memory complaints (SMCs) and is characterized by the self-perception of memory failures by the subject but without any functional limitation [5]. SMC may be an indicator of dementia risk, in fact, the preDIVA trial found a strong association between SMCs and dementia in a cohort of 3.454 community-dwelling older adults [6].

A recent meta-analysis suggests that people with SMCs have double the chance of developing dementia than those without any SMCs [7]. It is particularly noteworthy that in this study we found a difference between patients with SMCs that were firstly referred by an informer (externally referred SMCs) and those who self-referred (self-referral SMCs).

Stage 3 is MCI, and problems with memory, language, thinking, and judgment not related to aging are observed and identified by a neuropsychological study [8]. Nevertheless, in this phase there is no impact on the subject’s daily activities. The prevalence in Spain is 9.6% and it mainly affects women in the elderly [9]. The most characteristic profile of MCI due to AD usually manifests mainly in episodic memory, and there is characteristic profile of memory deficits, an amnestic syndrome of the hippocampal type, which is characterized by a loss of free and directed recall [10]. Patients perceive these memory losses and express their concerns.

Multiple behavioral, environmental, and genetic factors influence the different evolution of the disease [3]. Moreover, there are differences in the trajectories of cognitive aging, because some older people present considerable cerebral pathology without exhibiting concomitant decreases in cognition.

Therefore, the degree of brain pathology or brain damage does not appear to be, causally related to the clinical manifestation of the damage. This discrepancy can perhaps be explained by the theory of cognitive reserve and variations in cognitive aging whereby people with a greater cognitive reserve may be more resilient and thus, the presentation of the clinical signs of CI would be delayed in these subjects [11]. Cognitive function and its development is shown to be influenced by environmental and behavioral conditions both in adulthood and in the elderly. This occurs because even though neuronal plasticity is reduced by age, the possibility of creating new neurons, synaptic connections and new vessels is still maintained [12].

Cognitive reserve is determined by genetic and neurodevelopmental factors, although it may also vary depending on the environment and exposure to certain factors such as education and lifestyle [13]. Therefore, cognitive reserve is thought to be the result of an interaction between the patients’ genetics, environment, and experiences which, combined, would correspond to a set of skills that could actively compensate for the effects of the disease. Nonetheless, cognitive reserve is still a hypothetical construct that can only be measured indirectly by proxy indicators that are thought to represent it [14].

An active cognitive lifestyle seems to help protect individuals from cerebrovascular disease, while protective lifestyle factors such as social engagement and cognitive stimulation may contribute to increased cognitive reserve. Good social connections could also help increase cognitive reserve and can offer protection against declining cognitive function [11]. However, this latter association remains unclear because poor social relationships could be a consequence of cognitive decline rather than one of its causes. Nonetheless, having fewer social contacts, a smaller social network, and less engagement in social activities is associated with poorer cognitive function [15].

In this context, internet use has reinvented the ways in which we manage our social networks and relationships. Moreover, with the advent of smartphones, internet access has become portable. The widespread use of the internet has created new opportunities to learn and interact with society; even simple smartphone interactions through smartphone touchscreen interfaces produce neural changes in cortical regions associated with the sensory and motor processing of the hand and thumb [16].

Digital distractions on the internet seem to create a non-ideal environment for the refinement of higher cognitive functions during critical periods of brain development in children and adolescents [17]. However, the opposite may be true for older adults with CI, for whom the online environment may provide a new resource for positive cognitive stimulation [17].

To the best of our knowledge few studies have reported the implications of the use of the internet against CI in subjects over 50. For this reason, the first objective was to conduct a bibliographic review on the relationship between internet use and CI. The second objective was to detect associations between modifiable lifestyle factors and scores compatible with CI in people with SMCs.

## 2. Materials and Methods

This research is part of a multidisciplinary project formed by professionals from different disciplines, including community pharmacists, family practitioners, neurologists, psychologists, and mathematicians. We carried out a cross-sectional study to detect CI and potentially associated factors among the users of community pharmacies. The target population were adults aged over 50 years with SMC. In previous work, a decision tree obtained by machine learning techniques indicated that the presentation of a SMC was the variable most strongly associated with the detection of subjects with CI [18,19]. The idea was to maximize the selection process by focusing on those factors that imply a high probability of a positive screening test result. Therefore, SMC was used as an inclusion criteria in the study. In addition, SMC is an important marker in the prediction of cognitive deterioration [7].

### 2.1. Bibliographic Review

In order to assess the possible role of internet use in relation to CI in an older population group, a thorough literature review was conducted in PubMed and Web of Science databases. The keywords used and the inclusion and exclusion criteria are available in Table 1.

### 2.2. Patient Recruitment

The study was carried out in 19 community pharmacies located in the Valencian region (Spain), from September 2018 to January 2020. The inclusion criteria for this work were age over 50 years, having a SMC, and provision of informed consent to participate in the study. As shown in Figure 1, 502 people were initially screened, and 497 individuals met the inclusion criteria. Five subjects were not included because they had a diagnosis of dementia or a severe physical or sensory deficit—the exclusion criteria in this study. The use of medications indicated for dementia was the main criteria to exclude demented people.

Trained community pharmacists by the neurologist of our medical team performed active screenings to detect patient’s compatible with cognitive impairment (PCCI) in the individuals aged over 50 who visited their pharmacies.

Community pharmacists publicized the study using informational posters. During daily dispensing routine, pharmacists identified customers with SMCs (via patient self-referral, indirect questioning, or external referral), as the presentation of feelings of depression, an increased drowsiness, altered object recognition or language use, and difficulty performing complex activities such as using public transport, managing money, and/or following prescribed medical treatments. Individuals identified with SMC were informed about the study and were asked for their consent to participation. Identified patients were cited for a personal face-to-face interview, each interview lasted on average 50 min.

SMCs were either identified based on self-referral or an external referral. The former was identified if they answered yes to the any of following questions: “have you noted any memory loss?”, “have you noted any loss of object recognition?”, or “has your language use altered?”. The questions regarding the SMC were agreed with the neurologist of the research team. External-referral SMC was identified when a subjective complaint had been reported by someone other than the patient (e.g., a relative, pharmacist, or family doctor).

### 2.3. Cognitive Impairment Assessment

Subjects who presented these signs of SMC were invited to join the project, with a view to maximizing the future clinical diagnosis of patients with a potential CI. Among those we screened, three validated cognitive tests were completed. Few tests have high sensitivity and specificity simultaneously. In these cases, one can increase either the sensitivity or the specificity by combining two additional tests. These were the Memory Impairment Screen (MIS) test [20], Spanish version of Short Portable Mental State Questionnaire (SPMSQ) [21], and Semantic Verbal Fluency (SVF) [22,23] test, which were all selected after consultation with the Valencian Society of Neurology.

#### 2.3.1. Memory Impairment Screen

The MIS is a short memory disorder test that uses both the free and selectively facilitated recall of four words scoring on a 0–8 range. The optimal cut-off score is at ≤4 points, which corresponds to a sensitivity for dementia of 74% with a specificity of 96%, while for AD these results are 86% and 96%, respectively. Several validation studies have shown acceptable results for CI. For controlled learning and to ensure attention the MIS was used to induce specific semantic processing and optimize encoding specificity to improve detection of dementia. The MIS also presents a satisfactory correlation with the hippocampal and entorhinal volumetric measurements [20].

#### 2.3.2. Short Portable Mental State Questionnaire (Spanish Version)

This test assesses short-term memory, orientation, information about daily events, and ability to do serious mathematical work. Items include tasks on orientation (“What is the date today?”), memory (“What was your mother’s maiden name?”) and attention (“Subtract 3 from 20 and keep subtracting 3 from each new number, all the way down”). Thus, the cognitive scores ranged from 0 to 10 errors, with lower values which reflect better cognitive performance. The Spanish version of SPMSQ has a cut-off score of three errors; however, for illiterate people the cut-off score is four errors. It has a sensitivity of 85.7% and a specificity of 79.3% for the detection of CI [21].

#### 2.3.3. Semantic Verbal Fluency

The SVF test consists of a simple task of remembering words from a category, in this case animals. The researcher records the number of correct words cited by the patient for one minute. The recommended cut-off score for this test is 10 words, which corresponds to a sensitivity and specificity of 74% and 80% for the detection of CI, respectively [22,23].

Individuals with at least one positive test were classified as PCCI. These patients were referred to primary care with a letter including tests scores to obtain an early diagnosis [24].

### 2.4. Data Collection

We designed a specific questionnaire to collect information about possible variables associated with PCCI, which included as many demographic and lifestyle variables possibly related to CI as we could find in the literature [25].

All the variables were obtained by means of a face-to-face interview, which in our opinion facilitates greater objectivity. The qualitative variables collected were gender, age (50–59, 60–69, 70–79, ≥80), history of dementia (i.e., having family members with dementia), SMC (external referral or self-referral), educational level (preprimary, primary, secondary, or tertiary), marital status (married, separate, single, or widowed), feelings of depression and daily internet usage. The variable internet use was dichotomous, and it was referred to an active daily usage of the internet via electronic devices (computers, mobiles, or tablets).

Meanwhile, the quantitative variables were collected in number of hours and were divided into daily or weekly. The daily variables were daytime sleep and night sleep, memory training (specific exercises to enhance memory), doing hobbies (namely board games) and watching television. In the same way, the weekly variables were also collected in hours, multiplying the number of days and the average number of hours of the activity performed. These were physical exercise, reading and practising cognitively stimulating hobbies (particularly playing instruments or painting).

### 2.5. Sample Size Calculation

We used a sample size that was larger than necessary (385 people) to estimate the prevalence of PCCI and a SMC in the selected cohort. At minimum (in the least statistically favorable situation), we assumed this proportion would be 50% with an accuracy of 10% and a confidence level of 95%.

### 2.6. Statistical Treatment of the Data

All the patient information was collected in a hand-written medical history, which was used by a one researcher to populate and update a database in Excel software. The statistical analysis of the data was completed using R software (R Core Team, 2017, 2014). The prevalence of PCCI and a SMC was estimated with 95% confidence. We analyzed the association between patient profile variables, living and intellectual habits, and obtaining scores compatible with CI, as well as the association between these profile variables and habits.

To determine the association between qualitative variables, Chi-squared and Fisher exact tests were used. The association with respect to quantitative variables was analyzed with Pearson correlation coefficient tests and Student *t*-tests for independent samples, ANOVA and Kruskal–Wallis tests. Finally, to determine the protective or risk effect of the variables and to quantify their effect, both univariate and multivariate logistic regression models were adjusted to obtain estimates of the odds ratios (ORs) with 95% confidence.

### 2.7. Ethical Approval

The study was approved by the Research Ethics Committee at the Universidad CEU Cardenal Herrera (approval no. CEI18/027) and by the drugs Research Ethics Committee at Arnau de Vilanova Hospital (MOR-ROY-2018-013). In accordance with the Declaration of Helsinki, all the participants gave their written informed consent to participation.

## 3. Results

### 3.1. Bibliographic Review

Based on the search criteria, four longitudinal studies and one quasi-experimental study [26,27,28,29,30] have linked internet use to CI in subjects over 50 years of age. The longitudinal studies have had a follow-up time between 4 and 10 years and the samples have ranged from 897 to 8238 participants. The conclusions reached and information from the studies are shown in Table 2.

### 3.2. Demographic Characteristics of Individuals with a Subjective Memory Complaint

The following results are based on a sample of 497 subjects aged over 50 years with SMC who were screened at one of 19 collaborating community pharmacies. We also examined whether there were any differences between subjects that had self-referred their SMC versus those who had been externally referred. This classification was one of several other non-modifiable characteristics, including sex and age that defined the patient profiles.

As shown in Table 3, self-referred SMC was significantly related to having feelings of depression (OR = 2.24, 95% CI [1.34, 3.90]), taking anxiolytics or antidepressants (OR = 1.93, 95% CI [1.23, 3.05]), and being female (OR = 1.83, 95% CI [1.15, 2.88]). Conversely, no significant differences were observed for self-referred or externally referred SMCs in relation to age, family history of CIs, education level, or marital status. Unfortunately, no information on SMC or drug use was available for 5 and 19 patients, respectively.

### 3.3. Patient Scores on the Cognitive Tests

After screening, we divided the individuals based on whether their score on any of the three cognitive tests was compatible with CI (PCCI); scores were counted as incompatible with CI when all three tests had a negative result (No PCCI) (Table 4). This binary classification was a variable of interest in our research because it identified patients indicated for further neurological examination. As shown in Table 4, 30.8% (*n* = 153) of the study participants obtained scores compatible with CI. Thus, with 95% confidence, we can say that between 26.9% and 35.0% of subjects in the general population with a SMC and aged over 50 would also obtain scores compatible with CI.

### 3.4. Qualitative Variables

We then investigated the association between scores compatible with CI and the characteristics that defined the patient profile (Table 5), or with variables that defined living and intellectual habits (Table 6 and Table 7). The ORs were estimated with 95% confidence for any associations found with univariate logistic regression or with respect to the reference category (indicated by ‘1’) for each qualitative variable category.

There were significant associations between the SMC category and sex, depression and taking both anxiolytics and antidepressants. Furthermore, patient age, SMC type, educational level, marital status, and daily internet use were associated with PCCI. Specifically, subjects aged 70–79 years had 2.24–10.33-fold increased odds of obtaining CI-compatible scores compared to participants aged 50–59 years, and this was the characteristic that most strongly determined the patient profile. Furthermore, the OR for self-referral ranged from 1.35 to 3.90 and was almost 4 times higher than for externally referred individuals (Table 5).

Having a higher level of education (greater cognitive reserve) decreased the odds of scores consistent with CI by 0.09 to 0.63-fold (i.e., a 37% to 91% reduction in the reference odds) compared to subjects with a primary education only. Marital status may also explain scores compatible with suffering CI. For example, being widowed (i.e., generally more socially isolated) increased the odds of a score compatible with CI by between 1.5 and 3.77-fold compared to married individuals.

Interestingly, daily internet use was related to our variable of interest, with an OR of 0.14 to 0.33, meaning that this habit decreased the odds of obtaining scores compatible with CI by 67% to 86% with respect to participants who were not internet users (Table 5).

### 3.5. Quantitative Variables

As Table 6 shows, each extra hour of nightly sleep increased the odds of obtaining scores compatible with CI by between 1.02 and 1.28 times. Similarly, each additional hour of daily television viewing increased the risk by 1.08 to 1.30-fold.

However, each hour of reading per week decreased the odds by 0.82 to 0.93 times, equivalent to an odds reduction of 7% to 18%. We did not find any association between the likelihood of a score compatible with CI and the time subjects slept during the day or spent on hobbies, physical exercise, memory training, or pastimes.

Thus, based on this preliminary analysis, we deduced that eight characteristics related to the patient profile and habits were individually associated with CI scores.

Next, we assess whether each of these characteristics maintains or modifies its association with obtaining scores compatible with CI, in the presence of the others, in a multivariate logistic model. Prior to this step, we thoroughly analyzed the associations between these covariates, to establish possible influences of some covariates on others.

This analysis is summarized in Figure 2, an adapted Ishikawa diagram, in which the 8 covariates are organized into profile or habit variables. Profile variables are classified into modifiable and non-modifiable variables. On the other hand, habit variables are classified into whether they provide a risk or protective association. The dashed arrows between the variables represent all the associations found.

### 3.6. Multivariate Logistic Regression Models of the Patient Profile and Each of the Significant Modifiable Life Habits

Table 7 contains five multivariate logistic regression models, summarized in the five columns of the table. The first model has been carried out by including the four variables that define the patient profile. As can be seen, all the variables maintain the effect obtained with their respective univariate models shown in Table 5.

In the second model in Table 7, however, the hours of nighttime sleep lost its significance with respect to the probability of obtaining scores compatible with CI. This is probably explained by the relationship between hours of night-time sleep and age, as shown in Figure 2. In the third model, the reading habit with the presence of the profile variables maintained its significant protective effect. For every hour-spent reading per week, the odds of obtaining a score compatible with CI were reduced by 4% to 16%, compared to 7–18% obtained with the univariate model in Table 6.

The fourth model shows, the hours of TV watching maintained its risk effect in patients with CI scores such that each hour of daily TV increased the odds by 1% to 27% compared to an increase of 8% to 30% in the univariate model in Table 6. Furthermore, in the fifth model, internet and mobile device daily use maintained their protective effects, reducing the odds by 5% to 67% compared to non-users, versus a 67% to 86% reduction obtained with the univariate model. Of note, in models 3 and the estimations of the ORs for each habit based on the 95% confidence intervals was practically identical to those obtained in the univariate models. The fifth model, the profile variables reduce the effect that daily internet use had on suggestive CI scores compared to the univariate model in Table 5. Finally, the effects of the profile variables also maintained stable effects in each of the multivariate models.

## 4. Discussion

Lifestyle factors and cognitive stimulation have also been found to improve cognition and determine the individual risk of dementia [31]. Dementia is typically a disease of old age [1], and our results agree, as patients older than 70 years old have been associated with a higher probability of results compatible with CI. We found two lifestyle habits in that were associated with increased risk of CI: hours of nigh-time sleep and TV consumption. Conversely, cognitive stimulation variables such as reading, and daily internet use have been associated with a lower risk of CI.

Specifically, participants who were daily internet consumers (67–86%) and who usually read (7–18%) appeared to be at lower risk against scores indicating CI. In contrast, watching TV (8–30%) and additional hours of sleep (2–28%) were related to scores compatible with CI. Results which coincide with previous reports that excessive nocturnal sleep [32], and TV consumption [33,34] are related to the increased incidence of CI. However, the hours of night-time sleep lose their significance in the presence of the other variables, mainly age, which leads us to believe that in our sample this variable is not very influential.

Several studies have linked reading habits as a protective factor against CI [35,36]. In fact, reading increases brain connectivity in areas related to language and sensory regions [37]. Conversely, spending extended periods watching TV favors thickening of the frontal lobe and contributes to a decrease in verbal reasoning ability [38]. This may be because the way we interact with books and TV is quite different. While TV is designed to be passive and does not require great effort, reading books requires some concentration and reflection.

Conversely however, the role of internet use is more controversial. This is because it seems to affect differently according to the age of the patients in whom this factor has been examined. First, a failure of response inhibition and dysfunction within the inhibitory control network has been reported in young adult males with internet gaming disorder [39]. Similarly, in adults aged 18–30, problematic internet use led to major health problems [40]. Moreover, addictive internet use by young people with an average age of 22 years old was associated with increased functional connectivity density in the hyperactive impulsive habit system [41]. Finally, during critical developmental stages in children and adolescents, frequent internet use has been directly linked to decreased verbal intelligence and lower grey matter volume [42].

On the other hand, the opposite effect has been found in older adults (≥50 years) experiencing cognitive decline, as can be seen in the results obtained in the bibliographic review on this factor. Four longitudinal studies with a follow-up of 4 to 10 years have linked internet use to a decreased risk of cognitive impairment in patients over 50 years old [26,28,29,30]. Sensitivity analyses of the English Longitudinal Study of Ageing (ELSA) showed a reduction in the risk of developing dementia of between 43% and 58% [26]. However, a recent study observed this protective effect exclusively in men [29], which is a field of future research. Conversely, this effect of cognitive improvement does not only seem to be observed in patients with CI. Because a quasi-experimental study found that internet learning significantly improved episodic and visuospatial memory in patients without reported chronic diseases [27].

In our sample, daily internet use was the cognitive stimulation habit that was associated with the lowest risk of cognitive decline, over daily reading, or even the highest educational level (tertiary) in univariate logistic models. Furthermore, this significant effect was maintained in the presence of the other variables. This data opens the door to considering internet use as a cognitive stimulation variable in subjects over 50 years of age that may help to reduce the risk of CI in some cases.

Regarding SMC, it has been suggested as a preclinical marker of dementia risk in individuals with positive biomarkers [4,43]. This factor has been associated with a higher incidence of dementia compared to controls in longitudinal studies [7,44]. Specifically, individuals with self-referred SMCs represented 77.5% (the remaining 21.5% corresponds to patients with externally referred SMC) of our population and this factor was significantly related to feeling depressed and the consumption of anxiolytics or antidepressants, and the female sex. It would mean that in case of women reporting SMCs, they would be at an increased risk of showing compatible scores with CI than women in the general population. Forty-four percent of the people with a SMC in our study were taking anxiolytics or antidepressants. Anxiety and depression have long been associated with an increased risk of CI. This appears to be due to increased serum cortisol levels, due to hyperactivation of the hypothalamic-pituitary-adrenal axis [45]. These elevated cortisol levels may cause neuronal damage in the hippocampus, which could explain a possible link between these pathologies and CI [46].

Moreover, some authors have identified a quantitative relation between suffering anxiety or depression and SMCs [47]. This study supports our finding: in samples with SMCs the use of anxiolytics or antidepressants is higher than usual. Thus, given that these subjects both self-referred SMCs and had scores compatible with CI, perhaps the combination of symptoms of depression and consumption of these drugs could be used to reveal even more patients with neurodegenerative diseases in the future.

Concerning marital status, it may have also been related to CI in our study, given that widowers had a 1.5 to 3.77-fold increased odds of having a CI-compatible score compared to married individuals. Moreover, adjusting for age, SMC, and educational level, we observed that being single increased the odds of a CI-compatible score by 1.56 to 12.19-fold compared to married subjects. These results agree with studies [15,48,49,50] that support the idea social isolation related to those who live alone is a risk for a decline in cognitive function, while being married can sometimes buffer the effects that low mood has on the onset of dementia.

Regarding cognitive inactivity, it due to isolation might also explain the impact that marital status has on CI, although feelings of impaired cognition could also cause individuals to avoid social contact [1]. In contrast, cognitive reserve protects against CI. In fact, not having completed primary school studies predicts CI [1] and in this present work, we confirmed that having completed tertiary level studies decreased the odds of obtaining CI-compatible scores by between 37% and 91%.

Multiple variables have been related to cognitive stimulation, and this in turn to a decrease in the risk of cognitive decline. However, in our opinion, knowing the factors that are associated with the appearance of the first symptoms of cognitive decline in a high-risk population may be useful for prevention strategies. Among the variables that were significantly associated with CI-compatibility, the older the patient, the greater the number of hours of TV they consumed and hours of sleep they had, and the lower their internet use and reading habit. This could explain the increased risk of CI in older adults, not just in relation to age, but also with lifestyles that do not stimulate brain activity. Likewise, self-referral of a SMC was associated with subjects with a greater TV consumption habit and who used the internet less.

We hypothesized that as a sample with a common denominator, subjective memory complaint, the prevalence of AD is likely to be higher than in the general population. Therefore, with a small number of strict controls, the differences between the groups are less accurate than in a normal case-control study, as expected. This memory complaint could be present in patients as the first clinical manifestation of brain damage caused by β amyloid deposition, pathologic tau, and neurodegeneration, which allows AD to be considered as a biological continuum [51].

This study is part of the CRIDECO Team research lines and aimed to detect associations between modifiable lifestyle factors and scores compatible with CI in subjects with SMC [52]. The prevalence of PCCI (30.8%) was higher than in previous studies (17.4%) [15], due to the introduction of SMC as an inclusion criteria in our sample [18,24]. This increase in screening efficiency maximizes the rate of possible CI cases.

The importance of our study, in our view, is that it addresses the early stages of cognitive decline, known as “subjective or subtle objective cognitive decline” [5]. For many patients with AD, this will be the first symptomatic stage of the disease [3,4,53]. For this reason, all patients will be followed to establish evolutionary trajectories from the early stages.

This work highlights the importance of designing programs for older adults (in the at-risk population or among those with a known early diagnosis of CI) that stimulate the brain with activities such as reading and internet use to help slow the evolution of these diseases. It is also important to point out that SMC self-referral should be considered an important factor in our healthcare system. To help us develop effective cognitive training programs, our future lines of work are aimed at obtaining more information from these patients during their follow-up. In which the influence of these factors can be observed in the long term and allow us to draw more solid conclusions.

Limitations: The term internet use was dichotomous, and we do not have the number of hours that each patient used the internet daily, during the ongoing follow-up of patients this information is being asked. Finally, a detailed analysis of internet consumption among these subjects could help us to adjust the specific protection range of this variable according to individual profiles. Despite these limitations, one of our strengths is the sample size, which exceeds that needed to estimate the prevalence of PCCI and to support multivariate models.

We plan in the future to elucidate the relationship between compatible or incompatible scores (with cognitive impairment in some of the tests) with a definitive neurological diagnosis. This will be necessary to develop cognitive training programs, to increase brain stimulation via the habit of reading and the use of the internet in the population at risk, with a view to slowing down the evolution of the disease.

## 5. Conclusions

In our sample, self-referral of a SMC was significantly associated with CI, and was more common among women, people who felt depressed, and those taking anxiolytics or antidepressants. Furthermore, older adults with a SMC were particularly at risk of CI because of their lifestyle habits. In this work, internet use and reading significantly reduced the odds of scores compatible with CI. Moreover, four longitudinal studies and one quasi-experimental study found internet use to be beneficial against CI in older subjects. Conversely, the lifestyle variables that were associated with increased risk of cognitive decline were the number of hours of TV viewing and night-time sleep. However, future studies will be needed to ensure that these associations are confirmed in the long term during the follow-up of these patients.

## Figures and Tables

**Figure 1 jpm-11-01366-f001:**
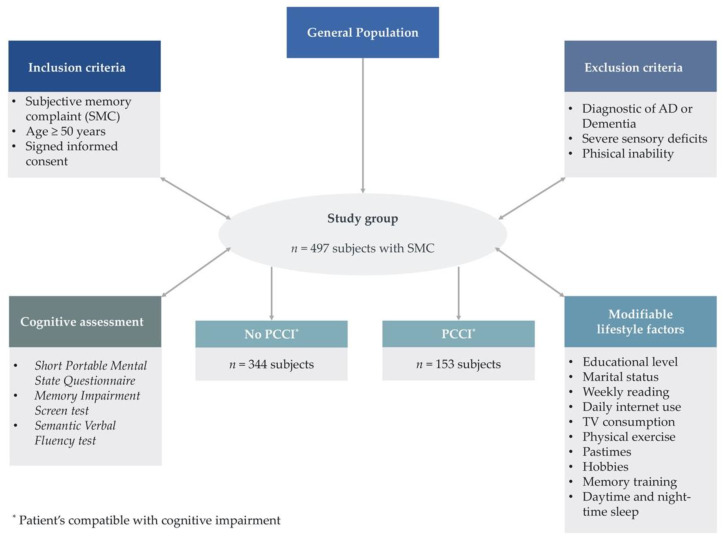
Diagram showing our analysis methodology. SMC: subjective memory complaint.

**Figure 2 jpm-11-01366-f002:**
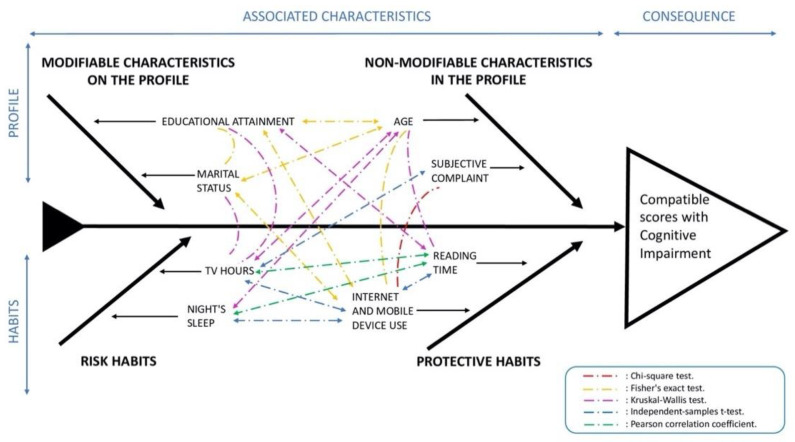
Characteristics associated with cognitive impairment scores (continuous arrows) and statistically significant associations between these characteristics (discontinuous arrows).

**Table 1 jpm-11-01366-t001:** Inclusion and exclusion criteria for the bibliographic review.

Inclusion Criteria	Exclusion Criteria
- Published in the last 5 years (2017–2021)	- Duplicated manuscripts
- Published in PubMed or Web of Science before November 2021	- Manuscripts not related to CI
- Population over 50 years old	- Screening using the title and abstract
- Language: English	- Manuscripts not specifically mentioning internet use
- Key words: “internet use” and “cognitive impairment” or “dementia”	- Manuscripts about molecular or non-commercialized drugs

**Table 2 jpm-11-01366-t002:** Bibliographic citations related to internet and their relationship against CI.

Study Type	Country (*N*)	Follow-Up	Sample Age	Relationship to Cognitive Impairment	Citation
Longitudinal	England; *N* = 8238 participants	10 years	>50 years	Internet use in individuals over 50 years of age was significantly associated with a 43–58% reduction in the risk of dementia.	d’Orsi et al., 2018 [26]
Quasi-experimental	Mexico; *N* = 27 participants	10 weeks	>60 years	Subjects who participated in the computer-based mental stimulation and internet learning program significantly improved their episodic memory and visuospatial processing compared to the control group.	Sánchez-Nieto et al., 2019 [27]
Longitudinal	Brazil; *N* = 1197 participants	4 years	>60 years	Significant association between continued internet use and cognitive status, with greater likelihood of cognitive gain and less cognitive decline.	Krug et al., 2019 [28]
Longitudinal	Switzerland; *N* = 897 participants	6 years	>65 years	Frequent internet use was associated with less subsequent cognitive decline. This effect was observed mainly in men.	Ihle et al., 2020 [29]
Longitudinal	England; *N* = 2530–3937 participants	8 years	>50 years	Internet use was associated with lower risk of cognitive impairment in the models used.	Williams et al., 2020 [30]

**Table 3 jpm-11-01366-t003:** The association between SMC and the patient profile variables.

Variable	Group	*n* (% Column)	SMC External Referral			SMC Self-Referral			*p*-Value	OR [95% CI]
			*n*	% Row	% Column	*n*	% Row	% Column		
Sex	Male	131 (26.6)	39	29.8	36.4	92	70.2	23.9	< 0.05 a	1
	Female	361 (73.4)	68	18.8	63.6	293	81.2	76.1		1.83 [1.15, 2.88] **
Depression	No	341 (69.3)	87	25.5	81.3	254	74.5	66.0	< 0.05 a	1
	Yes	151 (30.7)	20	13.2	18.7	131	86.8	34.0		2.24 [1.34, 3.90] **
Anxiolytics/Antidepressants	No	260 (54.5	70	26.9	66.7	190	73.1	50.9	< 0.05 a	1
	Yes	218 (45.6)	35	16.1	33.3	183	83.9	49.1		1.93 [1.23, 3.05] **

SMC: Subjective memory complaint; a: Chi-square test, univariate logistic models; **: *p*-value < 0.05.

**Table 4 jpm-11-01366-t004:** Patient classification based on whether the scores on three different screening tests.

Test		Total *n* (% Column)	No PCCI *n* (% Row)	PCCI *n* (% Row)	Total *n* (% Row)
SPMSQ	Normal	400 (80.4)	344 (86)	56 (14)	400 (100)
	Slightly impaired	74 (14.9)	0 (0)	74 (100)	74 (100)
	Moderately impaired	19 (3.8)	0 (0)	19 (100)	19 (100)
	Severely impaired	4 (0.8)	0 (0)	4 (100)	4 (100)
MIS Questionnaire	Normal	412 (82.9)	344 (83.5)	68 (16.5)	412 (100)
	Impaired	85 (17.1)	0 (0)	85 (100)	85 (100)
Verbal fluency Test	Normal	423 (85.1)	344 (81.3)	79 (18.7)	423 (100)
	Impaired	74 (14.9)	0 (0)	74 (100)	74 (100)
*N* positive test	Zero	344 (69.2)	344 (100)	0 (0)	344 (100)
	One	76 (15.3)	0 (0)	76 (100)	76 (100)
	Two	51 (10.3)	0 (0)	51 (100)	51 (100)
	Three	26 (5.2)	0 (0)	26 (100)	26 (100)
**Total**		497 (100)	344 (69.2)	153 (30.8)	497 (100)

PCCI: Patient’s compatible with cognitive impairment; SPMSQ: Short Portable Mental State Questionnaire; MIS: Memory Impairment Screen test.

**Table 5 jpm-11-01366-t005:** Qualitative variables on patient’s living.

Variable		Group	*n* (% Column)	No PCCI		PCCI		*p*-Value	OR [95% CI]
				*n*	% Row	*n*	% Row		
Non-Modifiable Characteristics	Sex	Female	364 (73.2)	250	68.7	114	31.3	0.742 a	
		Male	133 (26.8)	94	70.7	39	29.3		
	Age	50–59	74 (14.9)	65	87.8	9	12.2	<0.001 b	1
		60–69	155 (31.2)	130	83.9	25	16.1		1.39 [0.63, 3.30]
		70–79	191 (38.4)	117	61.3	74	38.7		4.57 [2.24, 10.33] ***
		≥80	75 (15.1)	30	40.0	45	60.0		10.83 [4.88, 26.34] ***
	Family history of dementia	No	315 (63.4)	210	66.7	105	33.3	0.130 a	
		Yes	181 (36.4)	133	73.5	48	26.5		
	SMC	External referral	107 (21.5)	87	81.3	20	18.7	0.001 a	1
		Self-referral	385 (77.5)	252	65.5	133	34.5		2.30 [1.35, 3.90] **
Modifiable Characteristics	Educational level	Preprimary	123 (24.7)	50	40.7	73	59.3		3.93 [2.46, 6.35] ***
		Primary	203 (40.8)	148	72.9	55	27.1		1
		Secondary	111 (22.3)	92	82.9	19	17.1		0.56 [0.30, 0.98] **
		Tertiary	57 (11.5)	52	91.2	5	8.8		0.26 [0.09, 0.63] **
	Marital status	Married	345 (69.4)	250	72.5	95	27.5	<0.001 b	1
		Separate	29 (5.8)	28	96.6	1	3.4		0.09 [0.005, 0.45] **
		Single	24 (4.8)	14	58.3	10	41.7		1.88 [0.79, 4.35]
		Widowed	99 (19.9)	52	52.5	47	47.5		2.38 [1.5, 3.77] ***
	Depression	No	345 (69.4)	247	71.6	98	28.4	0.092 a	
		Yes	152 (30.6)	97	63.8	55	36.2		
	Daily internet use	No	205 (41.2)	107	52.2	98	47.8	<0.001 a	1
		Yes	270 (54.3)	225	83.3	45	16.7		0.22 [0.14, 0.33] ***
Total			497 (100)	344	69.2	153	30.8		

PCCI: Patient’s compatible with cognitive impairment; SMC: subjective memory complaint. a: Chi-square test; b: Fisher exact test (univariate logistic models: **: *p*-value < 0.05; ***: *p*-value < 0.001.

**Table 6 jpm-11-01366-t006:** Association between PCCI or No PCCI and the quantitative variables on living and intellectual habits.

Variable	No PCCI			PCCI			*p*-Value	OR [95% CI]
	*n* (%)	Mean	*SD*	*n* (%)	Mean	*SD*		
Daytime sleep	343 (69)	0.41	0.58	153 (30.8)	0.47	0.7	0.325 c	
Night’s sleep	343 (69)	6.64	1.59	153 (30.8)	7.03	1.9	0.018 c	1.15 [1.02, 1.28] **
Hobbies	344 (69.2)	2.33	5.79	153 (30.8)	1.62	4.5	0.180 c	
Physical exercise	344 (69.2)	3.75	4.46	153 (30.8)	3.3	4.3	0.293 c	
Memory training	344 (69.2)	0.29	0.78	153 (30.8)	0.18	0.6	0.101 c	
Weekly reading	344 (69.2)	3.92	5.94	153 (30.8)	1.64	3.5	<0.001 c	0.88 [0.82, 0.93] ***
Pastimes	344 (69.2)	0.48	1.14	153 (30.8)	0.62	1.7	0.274 c	
Tv consumption	344 (69.2)	2.6	1.87	153 (30.8)	3.29	2.1	<0.001 c	1.18 [1.08, 1.30] ***

PCCI: patients compatible with cognitive impairment; TV: television. c: *t*-test for independent samples (univariate logistic models: **: *p*-value < 0.05; ***: *p*-value < 0.001).

**Table 7 jpm-11-01366-t007:** Multivariate logistic regression models on the patient profile and each of the significant life habits (*: *p*-value < 0.1; **: *p*-value < 0.05; ***: *p*-value < 0.001).

Variable				Profile	Profile + Night-Time Sleep	Profile + Reading	Profile + TV	Profile + Internet
				OR [95% CI]	OR [95% CI]	OR [95% CI]	OR [95% CI]	OR [95% CI]
Profile	Age	50–59		1	1	1	1	1
		60–69		1.17 [0.50, 2.96]	1.20 [0.51, 3.02]	1.22 [0.52, 3.09]	1.15 [0.49, 2.92]	1.04 [0.43, 2.66]
		70–79		2.68 [1.22, 6.49] **	2.67 [1.21, 6.48] **	2.81 [1.26, 6.87] **	2.55 [1.14, 6.20] **	2.08 [0.89, 5.26]
		≥80		4.62 [1.88, 12.24] **	4.51 [1.83, 11.97] **	5.23 [2.08, 14.16] ***	4.26 [1.71, 11.41] **	3.31 [1.24, 9.43] **
	SMC	External referral		1	1	1	1	1
		Self-referral		2.10 [1.17, 3.91] **	2.10 [1.17, 3.92] **	1.98 [1.09, 3.72] **	2.04 [1.13, 3.82] **	1.94 [1.05, 3.69] **
	Educational level	Preprimary		2.86 [1.17, 4.81] ***	2.89 [1.73, 4.87] ***	2.47 [1.46, 4.19] ***	2.98 [1.77, 5.03] ***	2.63 [1.53, 4.55] ***
		Primary		1	1	1	1	1
		Secondary		0.62 [0.32, 1.15]	0.64 [0.34, 1.20]	0.70 [0.36, 1.31]	0.66 [0.34, 1.23]	0.75 [0.38, 1.43]
		Tertiary		0.26 [0.08, 0.70] **	0.27 [0.08, 0.71] **	0.29 [0.08, 0.83] **	0.29 [0.09, 0.80] **	0.31 [0.09, 0.86] **
	Marital status	Married		1	1	1	1	1
		Separate		0.l8 [0.01, 0.93]	0.18 [0.01, 0.92]	0.17 [0.01, 0.85] *	0.16, [0.01, 0.82] *	0.20 [0.01, 1.01]
		Single		4.33 [1.56, 12.19] **	4.16 [1.49, 11.74] **	5.17 [1.79, 15.25] **	4.45 [1.58, 12.61] **	4.63 [1.57, 13.74] **
		Widowed		1.43 [0.84, 2.41]	1.42 [0.84, 2.41]	1.48 [0.87, 2.52]	1.40 [0.82, 2.36]	1.47 [0.85, 2.53]
Habits		Night-time sleep			1.07 [0.94, 1.21]			
		Reading				0.90 [0.84, 0.96] **		
		TV					1.13 [1.01, 1.27] **	
		Internet	No					1
			Yes					0.56 [0.33, 0.95] **

CI: cognitive impairment; SMC: subjective memory complaint. 95% confidence intervals for the odds ratio of patient profiles against each of the significant lifestyle habits (*: *p*-value < 0.1; **: *p*-value < 0.05; ***: *p*-value < 0.001).

## Data Availability

The data used for this study is available upon request.

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
