# Peer review of "Lifestyle Variables Such as Daily Internet Use, as Promising Protective Factors against Cognitive Impairment in Patients with Subjective Memory Complaints. Preliminary Results"

_jpm, 2021, doi:10.3390/jpm11121366_

Round 1

Reviewer 1 Report

Some of you findings seem weak and/or underpowered because you have so many independent variables. The variables in some are parsed down to a group of 7, making the confidence interval large and dependent on further subgroups of 3! I'd regroup some of these. Another example: your tertiary is broken down by 52 in one group and 5 in another.

The Ishikawa diagram is very complex.

You state each night time sleep was important. What was the range and scale? How were these subgroups broken down? You give a mean and SD, but the range is large. Why important but then not? Was there an interaction term? Most likely there is a missing variable.

The TV scores of 1-27% importance means possibly little to nothing (1%) or a lot at 27%. Unless one thinks 1% might be more than noise or measurement error.

I'd either break the groups down differently or issue a caution that some findings depend on a group of 3 people! For example, you claim an extra nights sleep increases odds 1.02-1.28, but the profile plus models suggests an interaction problem.

Very well thought out and methodical, but having so many variables must underpower your study, especially in a multivariate logistic regression analysis. It just leaves the reader believing because importance is there and then not when adding variables that there must be a different path which you do not explain.

Author Response

Please, see the attachment where we provide a point-by-point response.

Reviewer 2 Report

General comment

This is a timely research endeavor to identify tools and strategies for pronging brain health span in subjects at clinical risk for late-life cognitive impairment. Multi-dimensional interventions involving the individual medical/psychosocial/environmental (i.e., individual ecosystem) are desirable.

The clinical research value of this intellectual exercise is considerable. It complies with the necessity of tackling chronic, multi-factorial diseases before the onset or progress to stages when pharmacological interventions are needed, or therapeutic solutions are little likely to induce any meaningful effect.

However, while the present manuscript may reach the publication stage, such a positive evaluation is contingent on the authors’ willingness to fix conceptual shortcomings and potential pitfalls of inherent to the study design and analytical workplan.

Major points:

  1. The Introduction section fails to provide a comprehensive conceptual framework of subjective memory complaint and the potentially related biological processes that may influence cognitive training / environmental enrichment paradigms, e.g. AD clinical continuum.

  1. The statistical workplan and results presentation do not flesh out whether multiple comparisons correction was applied to rule out Type I bias. This is critical to draw clinical interpretations and assess the overall clinical meaningfulness of the piece.

the authors do not provide models assumption, diagnostics, and model fitting for the univariate / multivariate analysis. it is unclear whether and how features selection was performed

Moreover, I am concerned that the authors have relied too heavily on p-values and did not provide effect size information that would be helpful to balance statements when dealing with the relatively small population-number of tests balance of the present study.

  1. Lines 250 the authors state, “Machine learning techniques were used to select the candidates [15], who were included because they had a SMC and had consented in writing to participate in this study.”. it is unclear what this statement refers to. The term machine learning is not sufficient and a list of the commonly used algorithm shall be devised.

  1. The Discussion needs to be thoroughly revised to ensure overstatements since a few ORs look small

The Discussion lacks a sufficient argumentation on study limitations and how boost future clinical research on the matter in question. Is the absence of elaboration on study limitations an oversight or do the author think the study is free from any potential caveats?

Moreover, the inter-subject, inter-study variability of internet usage-related outcomes is lengthy and hard to follow.

Author Response

Please see the attachment where we provide a point by point response

Round 2

Reviewer 2 Report

The authors addressed all the points previously raised. The manuscript has been sufficiently improved for overall soundness and quality of narrative. Limitations have been acknowledged. I still feel that the introduction of SMC may be fine tuned further; though, this is not a concern.

Author Response

I would like to thank you for the exhaustive and useful improvements to the manuscript.

For better understanding we have reordered the paragraphs and we also included some additional information. We hope we have clarified the concept now.

Thank you very much again, for providing this helpful suggestion